# The Role of Tissue Geometry in Spinal Cord Regeneration

**DOI:** 10.3390/medicina58040542

**Published:** 2022-04-14

**Authors:** David B. Pettigrew, Niharika Singh, Sabarish Kirthivasan, Keith A. Crutcher

**Affiliations:** 1Department of Anatomy, Neuroscience and Health Sciences, University of Findlay, Findlay, OH 45840, USA; pettigrew@findlay.edu; 2School of Studies in Neuroscience, Jiwaji University, Gwalior 474011, India; niharikasingh1105@gmail.com; 3Department of Neurosurgery, University of Cincinnati, Cincinnati, OH 45267, USA; kirthish@mail.uc.edu

**Keywords:** spinal cord injury, tissue geometry, axonal regeneration, repair, myelin inhibitors

## Abstract

Unlike peripheral nerves, axonal regeneration is limited following injury to the spinal cord. While there may be reduced regenerative potential of injured neurons, the central nervous system (CNS) white matter environment appears to be more significant in limiting regrowth. Several factors may inhibit regeneration, and their neutralization can modestly enhance regrowth. However, most investigations have not considered the cytoarchitecture of spinal cord white matter. Several lines of investigation demonstrate that axonal regeneration is enhanced by maintaining, repairing, or reconstituting the parallel geometry of the spinal cord white matter. In this review, we focus on environmental factors that have been implicated as putative inhibitors of axonal regeneration and the evidence that their organization may be an important determinant in whether they inhibit or promote regeneration. Consideration of tissue geometry may be important for developing successful strategies to promote spinal cord regeneration.

## 1. Methods

The goal of this review is to provide a narrative overview of the concept that the organization of cellular elements within the spinal cord, herein referred to as the tissue geometry hypothesis, is a major determinant of the success or failure of axonal regeneration following injury. In order to narrow down the references to those most relevant to the hypothesis, a search of the peer-reviewed literature listed in PubMed was carried out using the following search terms (search results current as of 8 March 2022): spinal cord treatment (*n* = 111,051), peripheral nerve regeneration (*n* = 22,524), spinal cord white matter (*n* = 16,990), spinal cord repair (*n* = 15,007), spinal cord regeneration (*n* = 9532), spinal cord graft (*n* = 9465), spinal cord stem cell (*n* = 9028), glial scar spinal cord (*n* = 2355), spinal cord biomaterials (*n* = 1285), peripheral nerve scaffolds (*n* = 1712), spinal cord scaffolds (*n* = 1132), spinal cord myelin inhibitors (*n* = 998), spinal cord regeneration inhibitors (*n* = 944), spinal cord geometry (*n* = 394), and spinal cord fiber alignment (*n* = 156). To identify recent, relevant reviews of topics related to the hypothesis, the search term “review” was added to narrow down the list of papers within the areas of interest. Each of the co-authors were assigned the task of identifying the most relevant papers pertaining to one of four topics: inhibitors of axonal growth, neurogenesis and stem cells, artificial scaffolds (biomaterials), and clinical trials. The final selection of references was based on evidence for the cellular organization of factors that either inhibit or promote axonal regeneration.

## 2. The Problem

Spinal cord injury results in devastating morbidity and immense life-time health care costs. The immediate changes that occur following various types of injury to the spinal cord are well-described [1] and advances have been made in the acute care of these patients [2]. However, the long-term prognosis of such injuries remains poor. This is largely due to the inability of the injured tissue to restore the pathways and connections that have been damaged. Axonal regeneration following injury to spinal cord white matter is particularly limited. While in some cases, injury may involve complete transection of the spinal cord, far more common are contusion injuries in which the portions of the cord rostral and caudal to the injury site remain in continuity [3]. Nevertheless, axonal regeneration generally fails to occur across the injury site. Failure of regeneration is axiomatically assumed to underlie the etiology of the functional deficits associated with spinal cord injury. Consequently, establishing conditions that facilitate robust axonal regeneration and functional re-connectivity is generally assumed to be a prerequisite for successful treatment paradigms.

Despite the lack of axonal regeneration in spinal cord white matter, there is evidence that injured axons in the central nervous system (CNS) are capable of regeneration. Some of the earliest observations were made by Ramón y Cajal [4], who described “certain axons, few in number…that initiate regeneration…developing a small cone of growth”. This initial regeneration proceeds for as many as nine days, but “these sprouts penetrate very rarely into the wound itself (and) this neoformative action decreases or is arrested from the tenth to the fourteenth day”. There is abundant evidence that central axons are capable of regeneration if presented with an alternative environment. Grafts of non-CNS tissues such as peripheral nerve [5,6,7,8,9], striated muscle [8,10], iris [11,12], mitral valve [11], skin, tendon, thyroid and salivary glands [8] placed within the CNS have been shown to be invaded by central axons, although these results are often difficult to reproduce [8,10,11,13,14].

In contrast to those in spinal cord white matter, axons generally regenerate following peripheral nerve injury. Peripheral axonal regeneration is generally preceded by significant reorganization of the tissue milieu ahead of the advancing front of axons. Degeneration and clearance of distal axonal segments occurs [15], followed by reorganization of Schwann cells into longitudinal bands [16]. However, peripheral axonal regeneration is delayed in mutant mice in which there is a significant reduction in macrophage infiltration into the distal nerve stump [17,18,19] and delayed Wallerian degeneration [18,19]. These observations suggest that clearance and reorganization of the cytoarchitecture of the distal stump may be required before regeneration can occur.

Likewise, alterations to the cytoarchitecture of spinal cord white matter occur following injury. Within the injury zone, small perivascular hemorrhages appear within minutes along the longitudinal axis of the fiber tract [20]. Within 30 min, spaces appear between axons and their myelin sheaths [21,22]. Within one hour, the myelin sheaths disintegrate leaving unmyelinated segments of axons and scattered myelin debris within the injury zone [21]. Over the next few hours, axonal segments within the injury zone degenerate completely [4,20,23] and oligodendrocytes disappear [24]. Although macrophage invasion and the onset of debris clearance are rapid in the direct injury zone [24,25,26], myelin debris and lipid-laden cells can still be observed as late as 60 days following injury [24,26]. Within one week after injury, reactive astrogliosis occurs within and around the injury zone [27,28]. The resulting glial scar is characterized by “a dense mesh-work of hypertrophied astrocytic processes” [28] that form a complete border around the injury site between 1 week and 3 months post-injury [27]. Within the boundary of the injury site, the hypertrophied astrocytic processes appear anisomorphic in orientation but transition toward an orientation that is parallel with the longitudinal axis of the tract within the adjacent white matter [27]. 

Distal to the injury zone, a process similar to Wallerian degeneration occurs within CNS white matter [23,26,29,30] but appears slower or less robust than that in peripheral nerves. Compared with peripheral nerves, there is less macrophage invasion of injured optic nerve and spinal cord tracts [26,30,31]. The rates of Wallerian degeneration and pathway clearance are also comparatively reduced in the distal stump of central fiber tracts [26,29,30]. In fact, myelin debris can still be observed for as long as four months or three years following injury [23]. Thus, as appears to be the case in peripheral nerves of Wallerian degeneration-delayed mutant mice, slow clearance of debris and incomplete reorganization of normal white matter architecture distal to the injury site may pose a barrier to axonal regeneration.

Although the focus of this review is on the evidence for a role of tissue geometry, there are other factors that are relevant to any strategy for promoting functional recovery following spinal cord injury. For example, the decline in the intrinsic growth potential of CNS neurons with maturity appears to play an important role in limiting regeneration [32]. Treatments that target intracellular pathways involved in axonal transport, such as the mechanistic target of rapamycin (mTOR), phosphatase and tensin homolog (PTEN) and growth-associated protein 43 (GAP-43), offer potential therapeutic targets for enhancing axonal growth and functional recovery [33,34,35,36,37,38,39,40]. Similar evidence has been obtained for the use of neurotrophic factors, which can also contribute to greater regeneration and/or collateral sprouting [41,42,43,44]. Enhancing the intrinsic growth potential of axons, however, is unlikely to be sufficient if disruption of tissue organization prevents axonal navigation of the injury site.

## 3. Factors Associated with Glial Scars and Myelin Have Been Implicated in Growth Cone Collapse

The glial scar has been the focus of much research into the causes of regeneration failure [45,46,47,48,49,50,51]. In microtransplantation studies designed to reduce glial scarring, robust axonal growth occurs within white matter in vivo [27,52]. However, in cases in which more disruption of the host white matter and glial scarring occurred than intended, axonal growth halted at the boundary of the glial scar [27,52]. 

Much of the focus on glial scars as a barrier to regeneration within the CNS has sought to identify specific putative inhibitory molecules associated with the glial scar. Chondroitin sulfate proteoglycans upregulated in association with glial scars in the peri-injury zone have been shown to cause growth cone collapse or turning in vitro [28,53,54]. However, the extent to which the presence of these factors alone serves as a barrier to growth is difficult to assess. Within the glial scar, reactive astrocytes and putative inhibitors associated with them are not organized in parallel with the longitudinal axis of the fiber tract, unlike the astrocytes that characterize uninjured white matter [23]. Chondroitin sulfate proteoglycans are present in white matter distal to the injury site [55,56], which supports axonal growth, but there is relatively limited upregulation of these factors in distal white matter following injury [23] and these factors appear to be limited to a geometry that “surround axonal profiles” [55]. Microglia and macrophages at the glial scar site promote tissue repair and prevent infection, but also exhibit neurotoxic properties [57]. The fibrotic scar that lies in the cascade of reactive astrocytes has been referred to as an additional barrier to neuronal growth, since the dense network of fibroblasts produce a multitude of inhibitory factors to axonal growth [48,58].

The glial scar is now recognized to be more complicated than originally assumed in terms of its cellular composition and the expression of factors that either inhibit or enhance axonal growth [50,59]. In fact, Anderson et al. [60] provided evidence for a supportive, rather than inhibitory role of the scar in promoting regeneration. Furthermore, there is substantial evidence for positive roles for the glial scar, at least during acute phases of injury [48,51]. Studies have demonstrated that ablation of the astrocytic portion of the glial scar leads to neurological deficits and incomplete recovery, indicating that the presence of the astrocytic scar is important for functional recovery [61,62]. There is additional recent evidence that a glial scar exposed to an appropriate chemical environment may aid in functional recovery [51,63]. Other glial scar components such as microglia and oligodendrocytes may also have beneficial roles [64]. These findings suggest that the glial scar plays a much more complicated role in CNS injuries than previously assumed and that its removal may impede functional recovery.

Other putative inhibitors of axonal growth have been identified as normal components of myelin. As a result, the persistent presence of myelin debris distal to the injury site has been proposed to pose not only a physical barrier, but also to present molecular inhibitors of growth, thus depriving an advancing growth cone of a supportive growth environment. Several factors associated with myelin have been shown to induce growth cone collapse in tissue culture systems including Nogo, Myelin-Associated Glycoprotein (MAG), and Oligodendrocyte-Myelin Glycoprotein (OMgp) [65,66,67,68,69,70,71,72,73]. Lipid components of myelin have also been shown to be inhibitory to axonal growth [74,75]. Significant progress has been made in identifying the cellular mechanisms by which these factors lead to inhibition of neurite growth, as well as serving roles in regulating neuronal development and neuronal plasticity [76].

Much of the evidence supporting the designation of myelin-associated factors as putative inhibitors of growth come from experiments in which myelin, or its components, have been reconstituted as a substrate for cultured cells. For example, purified CNS myelin absorbed onto poly-lysine-coated culture dishes has been shown to inhibit cell attachment, neurite outgrowth and 3T3 fibroblast spreading [66,77,78], but not when myelin from MAG knockout mice is used [71]. Cultured oligodendrocytes and culture dishes coated with the myelin-associated protein Nogo have been shown to be non-permissive substrates for neuronal attachment, neurite outgrowth and 3T3 fibroblast spreading [65,66,72,78,79,80]. MAG-transfected Chinese Hamster Ovary cells were shown to inhibit neurite growth when used as substrates for postnatal cerebellar and adult sensory neurons [77,81]. The extracellular domain of MAG in solution inhibits neurite growth in vitro [73,82]. However, in these studies, myelin or myelin-associated factors are not presented to cultured cells or growing neurites in a manner mimicking their normal cytoarchitecture within the spinal cord. In healthy tissue, myelin and its associated factors are organized in parallel with the longitudinal axis of a white matter tract, a geometric organization that is obliterated in cases of homogeneous exposure to such factors, as is the case with reconstituted myelin absorbed onto culture dishes, cultured cells expressing myelin-associated factors, or myelin-associated factors in solution. 

Deactivation of myelin-associated inhibitors has been shown to promote axonal regeneration in systems that more closely resemble their normal organization. Monoclonal antibodies raised against Nogo can enhance axonal regeneration through white matter lesions sites in vivo [83,84,85,86,87,88,89,90] or in vitro [66,91,92]. Immunization with myelin, Nogo, or MAG has promoted corticospinal regeneration [93]. X-irradiation to block myelination has been shown to increase regeneration in crushed optic nerve [89]. Increased axonal regeneration occurs through lesions in the corticospinal tracts of MAG-knockout mice [69] and laser-inactivation of MAG increases regeneration of cultured retina and optic nerve [94]. Increased regeneration in CNS white matter has, likewise, been observed in mice in which various isoforms of Nogo have been knocked out [95,96], although these experiments have produced variable results [97,98,99].

Other studies have targeted the Nogo Receptor, which binds all three known myelin-associated inhibitor proteins [100,101,102]. Immunization with recombinant Nogo Receptor promotes regeneration in the spinal cord [61]. Transfection of retinal ganglion cells with a dominant negative form of Nogo Receptor promoted optic nerve regeneration in vivo [103]. Treatment with a competitive antagonist of the Nogo Receptor or the ectodomain of the Nogo Receptor in soluble form promotes regeneration of rubrospinal, raphespinal, and corticospinal fibers, often bypassing the lesion through adjacent gray matter [102,104,105,106,107]. Overall, these studies support the concept that to achieve optimal axonal regeneration in CNS white matter, the effects of myelin-associated inhibitors must be blocked. However, there are data supporting the concept that myelin-associated inhibitors are not entirely antagonistic to axonal regeneration and functional recovery. 

## 4. Intact White Matter Can Support Axonal Growth

Despite the presence of myelin-associated factors that induce growth cone collapse, there is clear evidence that CNS white matter supports axonal growth in cases where the cytoarchitecture remains intact. Systematic studies and serendipitous observations have been made of axonal growth over long distances in vivo within CNS white matter from transplanted embryonic neurons [27,52,108,109,110,111,112,113,114,115,116] or postnatal neurons [27,52,117].

The use of fresh-frozen tissue sections as substrates for neuronal culture has also provided evidence that white matter can support neurite growth. The technique, often referred to as cryoculture, has shown a variety of results, depending on the culture conditions [118,119,120,121,122,123,124,125]. When successful outgrowth from neurons cultured on cryostat sections of white matter occurs, it is generally directed in parallel with the longitudinal axis of the fiber tract but is limited on transverse sections through the same tissue [126,127]. Furthermore, on sections of the corpus callosum from myelin-deficient rats, the pattern of axonal growth is not parallel with the fiber tract and is morphologically indistinguishable from that on adjacent gray matter [127]. These observations are most consistent with the hypothesis that it is the spatial organization of the tissue components that determines whether axonal growth is successful. 

A direct test of this hypothesis was carried out in which the spinal cord was removed, crushed, and then immediately frozen to prevent glial scarring or other post-injury changes. Longitudinal cryostat sections through the crush site were then used as substrates on which primary neurons were cultured. Neurite growth occurred in parallel with the longitudinal axis of the tract on the uninjured areas of the white matter but did not extend across the crush site [126]. In contrast, the gray matter within the same injury site supported growth. These results demonstrate that disruption of the white matter tissue geometry is sufficient to inhibit growth, presumably due to the disorganized distribution of myelin and its associated inhibitors, that otherwise occurs on segments of white matter where the geometry is intact. 

To test the idea that similar disruption of the geometry of a peripheral nerve, whose myelin has inhibitory properties [71,128], might also impede axonal growth, additional experiments were conducted in which the sciatic nerve was removed, crushed, and then immediately frozen to prevent subsequent injury-induced changes. Longitudinal cryostat sections of the crushed nerve were then used as substrates to assess axonal growth. As found for the growth on longitudinal sections of crushed spinal cord, axons extended in parallel with the longitudinal axis of the nerve on segments where the nerve geometry is intact. However, these same axons did not cross injury sites [126]. Again, the assumption is that the disorganized myelin within the crush site served to inhibit growth through the region. The collective results support the conclusion that disruption of the tissue geometry, both in spinal cord white matter and in peripheral nerve tissues, is sufficient to create a largely non-permissive environment for axonal growth in the absence of scar formation. Beyond its potential role in contributing to axonal growth failure in injury, myelin-associated factors may normally serve to guide axonal growth in intact and developing tissue [70,87,127,129,130,131].

In newborn rats, X-irradiated locally to prevent myelination or treated with anti-Nogo antibodies, corticospinal axons regenerated over a larger cross-sectional area, intermixing with ascending axons running in the Fasciculi Gracilis and Cuneatus [70,87], suggesting that the absence of myelin eliminated guardrails that normally limit aberrant growth. Following spinal cord hemisection and combinatorial treatment including anti-Nogo antibodies and neurotrophin-3, regenerating axons appeared to cross the midline to bypass the lesion [132]. Treatment with anti-Nogo antibodies appeared to induce sprouting by the intact corticospinal tract across the midline, following a cervical spinal cord lesion [133] or an experimentally-induced ischemic stroke affecting the sensorimotor cortex [134]. Sprouting of the rubrospinal tract has been observed following treatment with anti-Nogo antibodies [135]. Similar collateral sprouting in the CNS has been observed in vivo following X-irradiation to suppress myelination [136,137], treatment with anti-Nogo antibodies [138,139] or targeted Nogo knockouts [140]. Following sciatic nerve transection, focal and temporary treatment with MAG reduced hyperinnervation and improved the accuracy of target reinnervation [141]. These studies suggest that myelin-associated factors may play an important role in guiding growth, constraining growth, and limiting collateral sprouting (which may or may not be beneficial).

Such a role for myelin-associated factors is supported by in vitro studies. Axons growing on patterned substrates consisting of alternating lanes of poly-lysine/laminin and myelin extend preferentially on poly-lysine/laminin but are oriented in the direction of the lanes, growing not on, but parallel to the myelin [142]. Treatment with MAG decreases axonal branching from dissociated dorsal root ganglion neurons [141]. Together, these in vivo and in vitro data support a role for putative inhibitors, normally found within white matter, in guiding or constraining axonal growth.

## 5. The Geometry Hypothesis

As reviewed above, there is evidence that both glial scars and myelin contain factors capable of limiting axonal growth. The tissue geometry hypothesis asserts that it is the spatial distribution of such factors that determines the success or failure of axonal growth. This hypothesis is shown schematically in the accompanying Figure 1. An example of white matter from the lateral funiculus of the spinal cord is illustrated with a growth cone advancing within a parallel bundle of uninjured myelinated axons. Myelin-associated inhibitors are shown as constraining growth to other cellular elements that are more permissive, such as unmyelinated axons. Following injury, disruption of the tissue geometry results in territory that no longer provides growth guidance through inhibitory and permissive factors. Reconstructing the lost tissue geometry, either through supportive biomaterials or grafts of cells capable of recreating the required architecture, is a central tenet of the geometry hypothesis.

If this hypothesis is correct, abortive growth may be inevitable where there is ubiquitous contact between growing axons and the growth cone-collapsing factors in the disrupted tissue. This is likely the case with the glial scar. Morphologically, the glial scar has the appearance of a dense anisomorphic boundary that encompasses the full perimeter of the injury site. Chondroitin sulfate proteoglycan-immunohistochemistry associated with glial scars has a corresponding morphology [27]. Chondroitin sulfate proteoglycans are, however, expressed away from the injury site and glial scar, albeit at lower levels, and more organized in a manner that “surrounds axonal profiles” [55], suggesting that these factors are organized in parallel with the longitudinal axis of the tract. Likewise, following injury to white matter, myelin debris, and presumably its associated factors, become disorganized and contact between advancing axons and myelin-associated factors becomes inevitable. Under the circumstances of the altered cytoarchitecture of the tissue close to the injury site, these factors become potent inhibitors of axonal growth.

Away from the injury site, however, the tissue geometry hypothesis posits that such factors remain relatively more organized, such that, rather than inhibiting axonal growth, they serve to guide it in parallel with the longitudinal axis of the fiber tract. As these factors appear to be arranged in parallel with the fiber tract, they may discourage growth perpendicular to the fiber tract. Organized in this fashion, the same factors that inhibit growth at the injury site may promote axonal growth by directing it efficiently down the long axis of the tract [130,131,143,144]. 

Furthermore, evidence from comparative studies, such as the axolotl’s phenomenal competency to regenerate the lost segments of its injured spinal cord [145,146], also supports the importance of tissue geometry. For example, prior to dedifferentiating, the SRY (sex determining region Y)-box 2-positive (SOX2+) neuronal stem cell pool aligns in a specific anterior-posterior direction [147]. This ensures that there is a specific spatial arrangement, i.e., anterior–posterior alignment, of the neural stem cell population before neurogenesis-aided regeneration occurs. Thus, although emphasis continues to be laid on the molecular prerequisites of the neural stem cells to facilitate regeneration, the role of tissue geometry is also important [148]. 

The idea that the spatial distribution of positive and negative factors could influence neurite growth was clearly articulated by Varon and Manthorpe [149]. In theory, the spatial distribution of both positive and negative factors within the tissue is potentially relevant. It is possible that the relative amount of both factors is distributed in a manner such that axons are directed by a path of least resistance. This could be due to the alignment of strong growth-promoting factors by themselves and/or the presence of negative factors oriented in parallel to the permissive factors, such as those postulated to be associated with myelin or possibly low levels of proteoglycans associated with fibrous astrocytes. As noted above, the net result is that the putative inhibitors of axonal growth normally serve to constrain the direction of growth or discourage collateral sprouting in directions perpendicular to the tract but not prevent it except where there is disruption of the normal tissue geometry. If so, it may be better to think of the putative growth “inhibitors” as factors that normally serve to constrain, direct, or even promote, axonal growth.

## 6. Reconstruction of Tissue Geometry to Promote Regeneration

Evidence supporting the importance of tissue geometry in successful regeneration comes both from tissue culture and animal models of CNS and PNS injury. As already noted, there is some evidence to indicate that grafting of a suitable tissue exhibiting the relevant geometry, such as a segment of peripheral nerve, can promote axonal regrowth in the CNS. However, the availability of such tissue grafts for clinical purposes is limited and must also address the issue of histocompatibility. An alternative approach is to use materials with low immunogenicity as a substrate to promote tissue integration and axonal regeneration (see below). Much of the initial work in this area has focused on the repair of peripheral nerve injuries, especially in cases where a large gap needs to be bridged. In fact, this continues to be an active area of investigation in which a variety of synthetic materials have been used [150,151]. The development of such materials generally assumes that reconstructing the normal tissue geometry of the peripheral nerve is essential for promoting effective regeneration.

The same concept is generally applicable to the injured spinal cord, where the goal is to reconstruct the relevant geometry in a way that best mimics the cytoarchitecture of the normal white matter tracts of the spinal cord. Attempts to develop suitable materials for this purpose include a wide variety of approaches ranging from cell-based therapies to the implantation of artificial scaffolds with specific architecture and combinations of both approaches [152,153,154,155,156,157,158,159,160]. Although most publications in material science do not cite the experimental evidence supporting the primary role of tissue geometry, the approaches that have been taken are largely based on the assumption that appropriate geometry is critical for promoting successful regeneration both in the PNS and CNS. Many different materials have been developed with the goal of providing a suitable matrix for repair of the injured spinal cord. Some of these have explicitly considered the relevant geometry of the white matter, whereas others have been designed with the goal of reconstituting the supportive cellular components of the tissue, while limiting the inflammatory and scar-promoting factors. 

One of the most productive areas of investigation in the use of biomaterials has been the development of geometry-based substrate materials to study neurite growth in tissue culture. Early work by Weiss and others [161,162] demonstrated the phenomenon of stereotropism (thigmotaxis), i.e., the guidance of neurites by topographical cues in tissue culture. This observation has since been extended to a wide variety of culture conditions and substrates. Some of the earliest studies to test whether aligned materials will support directional neurite growth were based on the use of novel fabrication methods, including microlithography and electrospinning. The latter approach permits the production of microfibers or nanofibers of various dimensions and alignment. For example, Yang et al. [163] provided some of the first evidence that aligned electro-spun polymer fibers promote better differentiation and neurite outgrowth of neural stem cells than fibers with random orientation. This difference between aligned and randomly-oriented fibers in supporting neurite outgrowth was also confirmed with primary neurons in vitro [164] and neurite growth was also found to occur from primary neurons that attached and extended neurites along bundles of carbon nanotubes [130]. 

One of the most compelling examples of the use of aligned materials to support neurite growth, both in vitro and in vivo, is the work of Hurtado et al. [165], which demonstrated the ability of aligned microfibers to support directional growth of neurites in a tissue culture system, as well as when implanted into an animal model of spinal cord injury. They further demonstrated that astrocytes also show aligned growth on such substrates in tissue culture, pointing to the possibility that cells normally involved in establishing a barrier to growth in the form of a scar might also be capable of promoting axonal growth when presented with the proper geometry.

There have been numerous subsequent studies that have used a wide variety of materials to explore what topographical or biochemical features are best suited to support directional growth of neurites as well as the alignment of glial cells that, in turn, would support such growth. This literature is extensive and has been reviewed by several investigators [154,156,157,166,167]. One of the overarching conclusions that can be gleaned from these studies is that aligned substrates, whatever means by which they are produced, exert compelling directional cues for growing neurites. 

Not only is geometry critical in promoting directional axonal growth, there is substantial evidence that geometry is also critical in directing cell differentiation. For example, aligned fibers have been reported to enhance the differentiation of neural stem cells along the Schwann cell lineage [168]. Thus, it seems reasonable to consider the use of relevant geometric cues to not only guide axonal growth but also to promote the development of cell morphologies that, in turn, can be used to contribute such guidance to axons.

Reconstructing the original geometry of the spinal cord, including the gray matter, requires reconstituting the neuronal circuitry that existed prior to the lesion. This includes the unique cytoarchitectural aspects of both the white and gray matter. Although most of this review is devoted to the question of how best to stimulate long-distance regeneration within the parallel geometry of white matter tracts, the goal of reconstructing local circuitry provides different challenges. To promote local connectivity, it is necessary to provide an architecture that mimics the high density of connections that normally characterize the gray matter. One approach is to use various grafts that provide the cellular elements normally found in this tissue, i.e., neurons, glia, and vascular elements. The use of neural stem cells within a supportive matrix is one such example (discussed below). Another approach is to provide a matrix that increases the amount of connectivity by reducing the tendency of fibers to fasciculate and form bundles of axons, the opposite of the goal with white matter. An interesting example of the latter approach is that of Usmani et al. [169], who used a 3D matrix of carbon nanotubes to inhibit fasciculation of axons in a tissue culture model to study the development of connectivity between spinal cord slices. They provided both histological and electrophysiological evidence for enhanced connectivity due to use of the 3D matrix.

In consideration of the role of local cytoarchitecture in promoting cell differentiation and appropriate connectivity, it may also be necessary to consider the intrinsic properties of neurons that can be modulated to promote regeneration. Modulation of transcription factors such as mammalian target of rapamycin (mTOR) promotes regeneration by upregulating the differentiation and migration of neural stem cells [170]. While such intrinsic mechanisms do promote regeneration, establishing functional contacts remains the goal. This is ensured by both intrinsic factors as well as topographical guidance cues within white matter tracts. In addition to appropriate differentiation of neural stem cells, restoring lost connectivity is equally important to obtain functional regeneration following injury to the spinal cord. 

The intrinsic growth patterns of different neuronal populations are also relevant. For example, interneurons do not normally extend long axons. Whether this reflects the primary role of the environment or the predisposition towards specific morphologies requires further investigation. One approach to this problem is to use cells in early stages of development, such as stem cells, to mimic the original cellular milieu. For this approach to succeed, it is necessary that not only the intrinsic developmental pathways of the stem cells are realized but also that the resulting tissue architecture is maintained. Such spatiotemporal control requires a balance to be maintained between intrinsic and extrinsic pathways that normally regulate development.

As noted above, the use of precise geometrical substrates to guide the growth of both neurons and glial cells provides the impetus for the construction of biomimetic scaffolds [160,171,172]. This concept has been extended to the use of co-aligned polystyrene nanofiber meshes to support the growth of neural stem cells, which have shown elongation and proliferation along the fiber axis of orientation [173]. The impact of geometry on stem cells has revealed influences on all cellular processes ranging from proliferation to differentiation. Manipulation of the spatial environment demonstrates an impact on stem cell fate. For example, stem cells placed in wells with circular boundaries lead to adipogenesis, while those in holly-shaped wells form osteocytes [174]. The two differently shaped wells reflect different mechanical stresses on the stem cells, simply due to the geometry of the wells because the holly-shaped wells exert greater mechanical stress than the circular wells. The underlying cellular mechanism remains to be elucidated but may involve Ras homolog family member A (RhoA) activity, because this member of the Rho family of GTPases controls stress fiber assembly [174]. These observations indicate that despite the occurrence of adult neurogenesis in specific neurogenic niches in the central nervous system, further investigation is required to answer questions that address the successful integration, implementation, and eventual translation of these processes into an effective application for the treatment of spinal cord injury.

## 7. Guidance Mechanisms

Both experimental and theoretical bases for the important role of tissue geometry in supporting axonal regeneration in the injured spinal cord is compelling. However, the specific mechanisms by which geometry serves to constrain or guide such growth remain to be elucidated. There are several mechanisms that have been proposed to regulate axonal growth. Ramon y Cajal [4] is usually credited with the initial proposal that axonal growth could be directed by diffusible chemical factors (chemotropism). Early work by Weiss demonstrated that neurite growth was also influenced by physical forces, i.e., stereotropism [161]. Subsequent work has provided evidence for several mechanisms that influence directional guidance of neurite outgrowth [175]. These include mechanical guidance (thigmotaxis), diffusible gradients (chemotaxis), substrate guidance (haptotaxis) and electric field guidance (electrotaxis). Recent work has also identified the influence of gradients of topological features on cellular migration (topotaxis) that could potentially influence neurite outgrowth [33]. 

There is evidence from studies of the development of pathways in the CNS that axons are guided by the organization of glial cells. For example, the development of the corpus callosum involves the guidance of axons by a transitory “sling” of glial cells organized longitudinally [176]. The development of the corticospinal pathway also implicates a critical role of channel guidance of the pioneer axons, followed by larger numbers of fibers that follow the pioneer axons [177]. Although the mechanisms operating during development may pertain to the problem of regeneration in the injured spinal cord, there are also clear differences. In some cases, the development of axonal projections relies on intermediate targets [178]. In addition, there is considerable evidence for reduced intrinsic growth potential as development proceeds, such that mature neurons are less capable of mounting a growth response [32]. Perhaps the most salient difference is that developing neurons encounter tissue that is itself developing so that the mechanisms guiding axonal growth are likely to be significantly different than those applying to the situation where regenerating axons encounter disrupted tissue. Nevertheless, the principles that apply to the development of neuronal pathways may have relevance to the mechanisms operating in the mature, injured spinal cord.

The role of geometry in the development and function of a wide variety of tissues has been an area of intensive investigation where tissue-specific properties determine the goals of the research. The use of recent bioprinting methods, for example, can be adapted for use with a variety of tissues [179]. The tissue geometry hypothesis, as applied to the problem of axonal regeneration, could theoretically include a variety of guidance mechanisms specific to the question of neuronal development. The results of the extensive studies involving the use of substrates with different geometric configurations without known chemical modification indicate that stereotropism (thigmotaxis) may be sufficient for obtaining directional growth. 

It is also possible that chemical factors that adhere to the substrate contribute guidance cues. For example, there is good evidence for the ability of substrate-bound factors to guide neurites (haptotaxis) [180,181]. However, the distinction between “mechanical” and “chemical” guidance may have little significance at the scale of the growth cone. What is clear from the available literature is that different surface topographies result in differences in both the rate and pattern of neurite outgrowth, even when no specific attempt has been made to alter the chemical composition of the surface. Another consideration that may have relevance to the development of prosthetic materials are the constraints imposed by intrinsic factors such as the energy costs and biomechanical limits associated with neurite turning and branching [182,183,184].

In the case of promoting aligned growth within spinal cord white matter, the relative role of positive and negative factors has not been established with certainty. In the case of the use of electrospun fibers, there is evidence that fiber diameter, inter-fiber spacing, and surface features influence the extent of outgrowth [185,186,187,188,189,190]. Thus, it is possible to envision a prosthetic material that relies exclusively on topological cues to promote successful regeneration. As far as we know, however, no attempt has been made to construct aligned materials that incorporate putative “inhibitors” of neurite growth to determine whether, as the geometry hypothesis predicts, such growth will not be reduced but may, in fact, be enhanced. This counter-intuitive proposal arises directly from the considerations outlined above.

## 8. Clinical Application of the Tissue Geometry Hypothesis

A variety of approaches based on the use of different materials are being pursued for clinical application. A recent clinical trial involving the use of a polymer matrix to promote spinal cord repair in a small sample of patients reported encouraging results [191]. Such material serves as a scaffold for cellular infiltration but does not otherwise attempt to mimic the normal geometry of the tissue. Similar, space-filling approaches have been used in pre-clinical models with the implantation of hydrogels [192,193,194] or a scaffold containing stem cells [195]. 

Based on the observations reviewed above, it is evident that the geometrical orientation of the tissue is of importance when considering the prospects of applying such results to the clinical problem of spinal cord regeneration. This is not to say that tissue geometry is the only factor that is relevant to the goal of promoting functional regeneration in the injured spinal cord. The cellular responses to injury, especially the inflammatory cascade, must be considered as well. Furthermore, accomplishing the goal of successful regeneration is unlikely to be sufficient by itself because any reconstituted pathways need to be incorporated in a manner that supports functional recovery. Regeneration of the damaged pathways may be necessary but is unlikely to be sufficient for such recovery. Furthermore, functional restoration depends on neurons restoring connectivity with appropriate downstream (for motor pathways) or upstream (for sensory pathways) targets. Encouraging axonal growth without appropriate connectivity could impede functional recovery if the regenerated pathways involve aberrant connections. 

In addition, the ability to encourage axons to grow in a manner that ultimately leads to functional changes needs to consider the reason that neurons grow axons in the first place. A large body of evidence supports the hypothesis that neurons extend processes to survive. That is, they seek the requisite trophic support provided by their target tissues. The need for trophic support has also been incorporated into some repair strategies by including growth factors or cells secreting such factors into the materials used to promote growth [41,42,196,197,198,199].

The vast literature on the question of how best to promote spinal cord regeneration, with the goal of establishing functional recovery, has only been touched on here. Numerous other reviews cover other aspects of this problem that are of equal importance [1,2,200,201,202,203,204,205,206,207]. The goal of this review has been to bring attention to the evidence for the primary role of tissue architecture in the development of strategies that seek to reconstruct the injured white matter. The specific role that tissue geometry plays in guiding axonal regeneration, as well as establishing functional connectivity, still remains to be elucidated in detail. The body of evidence, however, indicates that consideration of such geometry will be critical to the development of any strategy seeking to address the devastation that accompanies injuries to the spinal cord.

## Figures and Tables

**Figure 1 medicina-58-00542-f001:**
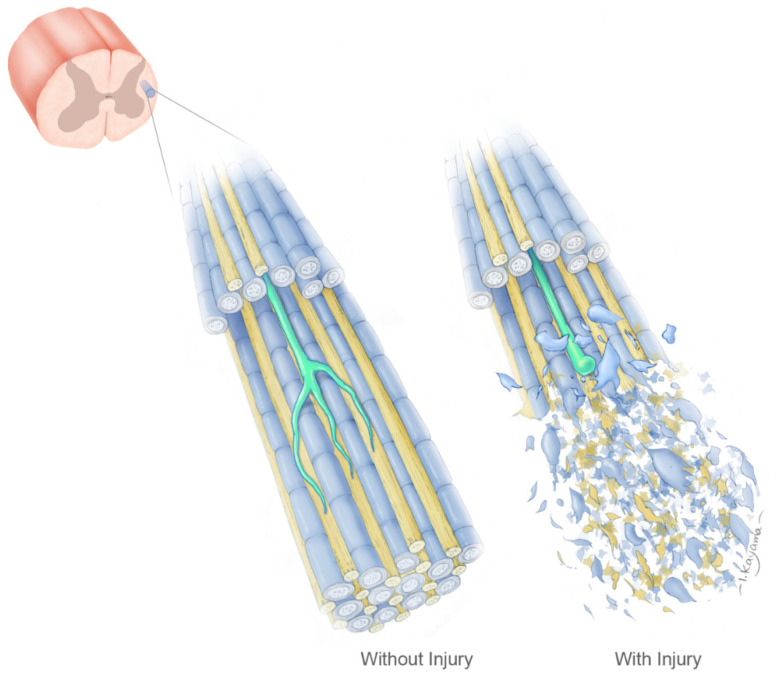
A representative sample of spinal cord white matter is shown in both intact (**left**) and injured (**right**) conditions. A growth cone (green) is shown advancing within a bundle of myelinated (blue) and unmyelinated (yellow) axons, although the latter could also represent any other permissive cellular elements within the tissue. The filopodia of the growth cone are shown advancing in association with the permissive elements but not along the myelinated axons, which present presumptive inhibitory factors. Following injury, the growth cone is no longer able to advance through the disrupted tissue due to loss of guidance from both the permissive and non-permissive tissue elements.

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
