# Peer review of "The Role of Tissue Geometry in Spinal Cord Regeneration"

_medicina, 2022, doi:10.3390/medicina58040542_

Round 1

Reviewer 1 Report

The authors have revised the manuscript , and it is acceptable in its current form

Reviewer 2 Report

Dear authors, thank you for allowing me to review this second version un your interesting manuscript. I found that all my previous comments have been addressed by authors, I have no further suggestions.

I found the quality of the paper as improved considerably.

Reviewer 3 Report

The authors have addressed my concerns and manuscript has been improved. 

This manuscript is a resubmission of an earlier submission. The following is a list of the peer review reports and author responses from that submission.

Round 1

Reviewer 1 Report

The following Topics should be revised in this review:

1- The search strategy for  review has not been described.

2-  The article needs a table to compare various references.

3- The article need illustration to better show the tissue geometry that has been described  and compare them.

Reviewer 2 Report

Dear authors,

This exciting manuscript focused on the description of the role of geometry in the regeneration of the spinal cord. 

Although the manuscript is well written and organized in its sections, I found it challenging to understand the design of the study you have conducted. In particular, I got only at the end of your manuscript (page 9, lines 419-422) that it is a review. I would kindly ask you to organize your paper as a (narrative or systematic?) review, with a methods section that discloses the search strategy and selection process of the papers you have selected/included in the review. 

The paper is more similar to a position paper than a literature review in this format.

Some minor comments are related to the extensive use of abbreviations and the lack of clarity about the existence of previous reviews on this topic, as reported on page 9, lines 418-419.

Reviewer 3 Report

The paper by Pettigrew et al. is a review about the role of tissue geometry in spinal cord regeneration. The topic is interesting and might be of interest for the field. However, bibliography does not reflect the current state of the art, mostly ignoring works since 2000, and the text lacks some internal coherence, in my opinion.

The whole idea of the review is to highlight the importance of tissue geometry and topology for regeneration, which is mainly supported by works with biomaterials. However, authors also introduce the effects of growth promoting or inhibiting factors, and even the idea of their spatial distribution, without properly discussing these effects vs those related to topology or geometry, and not showing studies demonstrating their involvement and interaction with geometry/topology. In addition, they only slightly discuss the great amount of data on this topic available from developmental studies.

Another point that they should reconsider is not only focusing on spinal cord white matter when intending to prove the effect of geometry, since a stronger regeneration is achieved in the gray matter of the spinal cord after some treatments (i.e. NPC transplants, PTEN deletion…), and that region does not show such a clear geometry or topology.

Authors make strong statements, such as “Experimental evidence supports the theory that regeneration failure in the CNS can be attributed more to the white matter environment than to reduced regenerative potential of central axons”, that may not be fully accurate. In this case, for example, there is a notable bibliography showing that modifications of intrinsic properties of neurons (i.e. interfering mTOR/PTEN signaling) strongly promote regeneration.

There is some lack of internal order that may result confounding for the reader, stating apparently contradictory ideas in different sections. For instance, in page 2, authors state that “Glial scarring within the injury zone and delayed Wallerian degeneration and reorganization of the cytoarchitecture of the distal stump in CNS white matter is widely hypothesized to pose a physical barrier to axonal regeneration”, ignoring the new works that challenge the traditional view of the glial scar as an inhibitor, and that differentiate among different types of glial cells (astrocytes, microglia) and other components of the lesion core/borders (laminins, CSPGs, fibroblasts, pericytes). However, several paragraphs in advance ,in the next page, this new view is partially acknowledged by the authors. Maybe a small review of the text may help to clarify the authors point.

Minor changes:

There are repeated references (42, 44), and some references not properly cited (91, 92)